# Association of depression and antidepressant therapy with antiretroviral therapy adherence and health-related quality of life in men who have sex with men

Yung-Feng Yen[1,2,3,4,5]*, Hsin-Hao Lai[1,6], Yen-Chun Kuo[2,7], Shang-Yih Chan[3,5,8], Lian-Yu Chen[9,10], Chu-Chieh Chen[3], Teng-Ho Wang[11], Chien Chun Wang[12], Marcelo Chen[13,14], Tsen-Fang Yen[15], Li-Lan Kuo[15], Shu-Ting Kuo[15], Pei-Hung Chuang[1]

1 Section of Infectious Diseases, Taipei City Hospital, Yangming Branch, Taipei, Taiwan, 2 Institute of Public Health, National Yang Ming Chiao Tung University, Taipei, Taiwan, 3 Department of Health Care Management, National Taipei University of Nursing and Health Sciences, Taipei, Taiwan, 4 Department of Education and Research, Taipei City Hospital, Taipei, Taiwan, 5 University of Taipei, Taipei, Taiwan, 6 School of Medicine, National Yang Ming Chiao Tung University, Taipei, Taiwan, 7 Department of Psychiatry, Taipei City Hospital, Linsen, Chinese Medicine, and Kunming Branch, Taipei, Taiwan, 8 Department of Internal Medicine, Taipei City Hospital, Yangming Branch, Taipei, Taiwan, 9 Kunming Prevention Center, Taipei City Hospital, Taipei, Taiwan, 10 Institute of Epidemiology and Preventive Medicine, National Taiwan University, Taipei, Taiwan, 11 Section of Infectious Diseases, Taipei City Hospital, Zhongxiao Branch, Taipei, Taiwan, 12 Section of Infectious Diseases, Taipei City Hospital, Linsen, Chinese Medicine, and Kunming Branch, Taipei, Taiwan, 13 Department of Urology, Mackay Memorial Hospital, Taipei, Taiwan, 14 Department of Cosmetic Applications and Management, Mackay Junior College of Medicine, Nursing and Management, Taipei, Taiwan, 15 Department of Nursing, Taipei City Hospital, Linsen, Chinese Medicine, and Kunming Branch, Taipei, Taiwan

☯ These authors contributed equally to this work.
* dam37@tpech.gov.tw

**Data Availability Statement:** All relevant data are within the paper and its Supporting information files.

## Abstract

UNAIDS' HIV treatment targets require that 90% of people living with HIV/AIDS (PLWHA) receiving antiretroviral treatment (ART) achieve viral suppression and 90% of people with viral suppression have good health-related quality of life (HRQOL). This study aimed to examine the association of depression and antidepressant therapy with ART adherence and HRQOL in HIV-infected men who have sex with men (MSM). From 2018 through 2020, HIV-infected MSMs were consecutively recruited (*N* = 565) for the evaluation of ART adherence and HRQOL at Taipei City Hospital HIV clinics. Non-adherence to ART was defined as a Medication Adherence Report Scale score of < 23. HRQOL in PLWHHA was evaluated using WHOQOL-BREF, Taiwan version. Overall, 14.0% had depression and 12.4% exhibited non-adherence to ART. The nonadherence proportion was 21.8% and 10.5% in depressed and nondepressed HIV-infected MSM, respectively. After adjusting for other covariates, depression was associated with a higher risk of nonadherence to ART (adjusted odds ratio = 2.02; 95% confidence interval: 1.02–4.00). Physical, psychological, social, and environmental HRQOL were significantly negatively associated with depression. Considering antidepressant therapy, ART nonadherence was significantly associated with depression without antidepressant therapy but not with antidepressant therapy. The depressed HIV-infected MSM without antidepressant therapy had worse psychological, social, and

**Funding:** This study was supported by two grants from the Ministry of Science and Technology, Taiwan (MOST108-2410-H-532-001) and the Department of Health, Taipei City Government, Taiwan (No. 10801-62-006). The study sponsors were not involved in the study design, the collection, analysis, or interpretation of the data, the writing of this report, or the decision to submit it for publication.

**Competing interests:** The authors have declared that no competing interests exist.

environmental HRQOL than those with antidepressant therapy. Our study suggests that depression is associated with poor ART adherence and HRQOL, particularly in those without antidepressant therapy. Adequate diagnosis and treatment of depression should be provided for PLWHA to improve their ART adherence and HRQOL.

## Introduction

As of 2019, there were 38 million people living with HIV/AIDS (PLWHA) worldwide [1]. With the widespread use of highly active antiretroviral therapy (ART), PLWHA are living longer [2] and have more neuropsychiatric comorbidities [3]. Depression is the most common neuropsychiatric comorbidity in PLWHA and it can occur in all phases of the infection [4]. However, depressive symptoms in PLWHA are often neglected in HIV care facilities [5].

In 2014, the Joint United Nations Programme on HIV/AIDS (UNAIDS) and World Health Organization announced the 90-90-90 UNAIDS targets to end the HIV epidemic; that is, that 90% of all PLWHA in a community or country are aware of their status, 90% of those aware have sustained ART, and 90% of those receiving ART achieve durable viral suppression [6]. To achieve viral suppression in PLWHA, adherence to ART plays an important role in determining the treatment success [7]. Paterson et al. previously reported that achieving and maintaining virological success in PLWHA requires an adherence rate of about 95% [8]. Poor adherence to ART in PLWHA not only increases the risk of death [9] but also results in the transmission of HIV to others in the absence of condom usage or pre-exposure prophylaxis [10].

Health-related quality of life (HRQOL) is a multidimensional construct that is concerned with the impact of health on individuals' level of functioning and the perception of their well-being in important areas of their life [11]. Since ART has significantly improved the survival of HIV-infected individuals, PLWHA are increasingly concerned not only with a treatment's ability to extend their life but also the quality of the life that they are able to lead. Therefore, UNAIDS has added a "fourth 90" to the prior three-pronged 90-90-90 targets to improve mental well-being in PLWHA, which ensures that 90% of PLWHA have good HRQOL and are linked to integrated health services [12–15].

In Taiwan, HIV infection was first detected in 1984 among men who have sex with men (MSM) [16]. Since then, the number of PLWHA in Taiwan has gradually increased. At the end of 2019, the number of PLWHA had reached 39,669, of which 65.0% were MSM [16]. Although free ART has been offered to all HIV-infected individuals since 1997 [17], it is estimated that around 80% of all HIV-infected patients in Taiwan receive ART [16], and 80% of those achieve undetectable levels of HIV [16].

Depression is not uncommon in PLWHA [18] and it has been associated with HIV disease progression and accelerated CD4+ cell decline [19, 20]. However, the adequate diagnosis and treatment of depression has not been routinely provided for PLWHA [4, 5]. Since UNAIDS' HIV testing and treatment targets require that 90% of PLWHA who receive ART should achieve durable viral suppression and that 90% of people with viral load suppression should have good HRQOL, it is imperative to identify the modifiable factors associated with poor ART adherence and HRQOL and to implement interventions to change these factors in order to improve ART adherence and HRQOL in PLWHA. Therefore, this study aimed to determine the association of depression and antidepressant therapy with ART adherence and HRQOL in HIV-infected MSM in Taiwan.

## Methods

### Study population and eligibility

This study consecutively recruited HIV-infected patients from Taipei City Hospital (TCH) HIV clinics, the largest HIV care center in Taiwan, between December 2018 and May 2020. As of the end of 2019, a total of 4,150 PLWHA are regularly followed-up at TCH HIV clinics. Subjects enrolled in our study were 18 years of age or older, were receiving ART, and had provided written informed consent. If a study participant agreed, the case manager interviewed the subject about their treatment adherence and HRQOL. Study participants who completed the survey were compensated with a coupon of US 3 dollars for their time.

Since the numbers of heterosexuals infected with HIV (n = 27) and PLWHA who were female who had sex with female (n = 1) are limited, this study only included PLWHA in the analysis who were men who had sex with men (n = 565) to determine the association of depression and antidepressant therapy with ART adherence and HRQOL. This study was approved by the Institutional Review Board of TCH (no. TCHIRB-10612120) and all interview with the study participants were performed in accordance with TCH IRB guidelines and regulations.

### Measurement of treatment adherence

This study used the Medication Adherence Report Scale (MARS-5) to evaluate the treatment adherence in PLWHA [21]. The MARS-5 has been used as a self-reported measure of adherence for patients with heart failure [22] and asthma [23], and it has been proven to have good reliability and validity [21]. The MARS-5 includes five common patterns of nonadherent behavior that respondents score on a five-point Likert scale (1 = always, 2 = often, 3 = sometimes, 4 = rarely, and 5 = never) (S1 Table). The first item of the MARS-5 questions PLWHA about unintentional nonadherence, whereas the other four items ask about intentional nonadherence [24]. The scores are summed and the total ranges from 5 to 25. Lower scores indicate lower self-reported adherence. Nonadherence to ART is defined as a MARS-5 score of < 23 [24].

### Assessment of HRQOL

The participants' HRQOL was evaluated using the Taiwanese version of the short form of the World Health Organization Quality of Life questionnaire (WHOQOL-BREF). The Taiwanese version includes the 26 original items of the WHOQOL-BREF [25] plus two culture-specific items that are relevant to Taiwan [26], that were proposed by patient and expert focus groups after a qualitative analysis of the recorded content. Based on the psychometric analyses, one culture-specific item that addresses "respect from others" was included in the social domain, and the other item that corresponds to "eating what one likes to eat" was included in the environmental domain. The scoring procedures, method of application, and reference time point (during the last 2 weeks) were the same as in the original WHOQOL-BREF [25]. The test statistic for the reliability of the Taiwanese version of the WHOQOL-BREF and the original measure ranges from 0.70 to 0.80 and the Cronbach's alpha is between 0.70 and 0.77 [26, 27]. The Taiwanese version of the WHOQOL-BREF measures four domains that include physical capacity (seven items), psychological well-being (six items), social relationships (four items), and environmental health (nine items). All items are rated on a five-point scale and a higher score indicates better HRQOL. Since each domain has a different number of items, the domain scores were calculated by multiplying the average of the scores of all items in the domain by

the same factor of 4. Therefore, each domain score had the same range, which was from 4 to 20 [28].

## Data collection

At the time of the study enrollment, consenting participants underwent a face-to-face interview that was administered by a trained case manager using a standardized questionnaire. This questionnaire was used to collect information about the participants' MARS-5 score, HRQOL, sociodemographic characteristics, depression, antidepressant therapy, and history of sexually-transmitted diseases. The sociodemographic characteristics included age, sex preference, income, unemployment, education, smoking, and alcohol use. Participants' history of sexually-transmitted diseases included syphilis, gonorrhea, and genital warts. The viral load and CD4 counts at the time of enrollment were collected from the participants' medical charts.

## Outcome variables

The primary outcomes of this study were adherence to ART and HRQOL. Adherence to ART was determined by the MARS-5 score. Nonadherence and good adherence to ART were defined as a MARS-5 score of $< 23$ and $\geq 23$, respectively [24]. HRQOL was evaluated using the Taiwanese version of the WHOQOL-BREF, with a higher score indicating better HRQOL [26].

## Statistical analyses

The demographic data of the study participants were analyzed (S2 Table). Continuous data were presented as the mean (standard deviation [SD]), and the two-sample $t$-test was used to compare groups. Categorical data were analyzed using Pearson's $\chi^2$ test, where appropriate.

Multivariate logistic regression was used to estimate the association of depression and antidepressant therapy with the ART adherence in PLWHA after adjusting for the participants' age, income, and history of sexually transmitted diseases. The variable with $p<0.05$ was defined as a significant factor that was associated with nonadherence to ART in the multivariate analysis. Adjusted odds ratios (AOR) with 95% confidence intervals (CI) were reported to show the strength and direction of these associations.

This study used linear regression to assess the univariate and multivariate associations of depression and antidepressant therapy with each HRQOL domain. All of the data management and analyses in this study were performed using the SAS 9.4 (SAS Institute, Cary, NC, USA) and IBM SPSS 19.0 (IBM Corp., Armonk, NY, USA) software packages.

# Results

## Participant selection

This study included 565 HIV-infected MSM who were evaluated for the adherence to ART and HRQOL between December 2018 and May 2020. After excluding those with missing data ($n = 8$), the remaining 557 HIV-infected MSM were included in the analysis. The overall mean (SD) age was 37.7 (8.5) years, and 14.0% of the participants had depression and 12.4% reported nonadherence to ART.

## Characteristics of depressed and nondepressed HIV-infected MSM

Table 1 shows the characteristics of the depressed and nondepressed HIV-infected MSM. Compared to the nondepressed HIV-infected MSM, the depressed HIV-infected MSM were more likely to report nonadherence to ART (21.8% vs 10.9%) and to have poor physical,

**Table 1. Characteristics of the HIV-infected MSM, by depression status.**

| Characteristics | Total = n = 557 | No. (%) of subjects[*] | | P value |
|---|---|---|---|---|
| | | HIV-infected MSM without depression = n = 479 | HIV-infected MSM with depression = n = 78 | |
| **Demographics** | | | | |
| Age, yr | | | | |
| Mean (SD) | 37.7 (8.5) | 37.7 (8.4) | 37.3 (9.0) | 0.716 |
| 15–39 | 338 (60.7) | 289 (60.3) | 49 (62.8) | 0.677 |
| ≥40 | 219 (39.3) | 190 (39.7) | 29 (37.2) | |
| Education level completed | | | | |
| ≤High school | 134 (24.1) | 109 (22.8) | 25 (32.1) | 0.075 |
| University or above | 423 (75.9) | 370 (77.2) | 53 (67.9) | |
| Income level | | | | |
| Low | 65 (11.7) | 45 (9.4) | 20 (25.6) | <.001 |
| Intermediate | 233 (41.8) | 199 (41.5) | 34 (43.6) | |
| High | 259 (46.5) | 235 (49.1) | 24 (30.8) | |
| Unemployment | | | | |
| No | 500 (89.8) | 432 (90.2) | 68 (87.2) | 0.416 |
| Yes | 57 (10.2) | 47 (9.8) | 10 (12.8) | |
| Any alcohol use | | | | |
| No | 304 (54.6) | 267 (55.7) | 37 (47.4) | 0.172 |
| Yes | 253 (45.4) | 212 (44.3) | 41 (52.6) | |
| Smoking | | | | |
| No | 341 (61.2) | 307 (64.1) | 34 (43.6) | 0.001 |
| Yes | 216 (38.8) | 172 (35.9) | 44 (56.4) | |
| Use of hypnotic drugs | | | | |
| No | 514 (92.3) | 449 (93.7) | 65 (83.3) | 0.001 |
| Yes | 43 (7.7) | 30 (6.3) | 13 (16.7) | |
| **History of sexually transmitted diseases** | | | | |
| History of syphilis | | | | |
| No | 245 (44.0) | 213 (44.5) | 32 (41.0) | 0.57 |
| Yes | 312 (56.0) | 266 (55.5) | 46 (59.0) | |
| History of gonorrhea infection | | | | |
| No | 497 (89.2) | 433 (90.4) | 64 (82.1) | 0.027 |
| Yes | 60 (10.8) | 46 (9.6) | 14 (17.9) | |
| History of warts | | | | |
| No | 455 (81.7) | 391 (81.6) | 64 (82.1) | 0.929 |
| Yes | 102 (18.3) | 88 (18.4) | 14 (17.9) | |
| **CD4 count, cells/mm³** | | | | |
| <350 | 74 (13.3) | 65 (13.6) | 9 (11.5) | 0.841 |
| 350–499 | 146 (26.2) | 124 (25.9) | 22 (28.2) | |
| ≥500 | 337 (60.5) | 290 (60.5) | 47 (60.3) | |
| **HIV-1 RNA, copies/ml** | | | | |
| HIV-1 RNA<40 | 500 (89.8) | 428 (89.4) | 72 (92.3) | 0.425 |
| HIV-1 RNA≥40 | 57 (10.2) | 51 (10.6) | 6 (7.7) | |
| **Treatment adherence to ART** | | | | |
| High adherence | 488 (87.6) | 427 (89.1) | 61 (78.2) | 0.007 |
| Non-adherence | 69 (12.4) | 52 (10.9) | 17 (21.8) | |
| **WHOQOL-BREF domain, mean (SD)** | | | | |
| Physical | 14.4 (2.5) | 14.7 (2.3) | 12.6 (2.6) | <.001 |

(*Continued*)

**Table 1.** (Continued)

| Characteristics | Total = n = 557 | No. (%) of subjects* | | P value |
| --- | --- | --- | --- | --- |
| | | HIV-infected MSM without depression = n = 479 | HIV-infected MSM with depression = n = 78 | |
| Psychological | 13.3 (2.9) | 13.6 (2.7) | 11.7 (3.1) | <.001 |
| Social | 13.3 (2.6) | 13.5 (2.4) | 11.9 (3.1) | <.001 |
| Environmental | 14.3 (2.3) | 14.5 (2.2) | 13.1 (2.6) | <.001 |

MSM = men who have sex with men; ART = highly active anti-retroviral therapy; SD = standard deviation; WHOQOL-BREF = short form of the World Health Organization Quality of Life questionnaire.

*Unless stated otherwise.

psychological, social, and environmental HRQOL. Moreover, the depressed HIV-infected MSM had a lower income and were more likely to smoke and use hypnotic drugs. In terms of the sexually transmitted diseases, the depressed HIV-infected MSM were more likely to have ever been infected with gonorrhea.

## Factors associated with nonadherence to antiretroviral therapy

Multivariate logistic regression was used to identify the independent risk factors for nonadherence to ART in HIV-infected MSM. After controlling for the demographic characteristics and other covariates, depression was found to be associated with a higher risk of nonadherence to ART (AOR = 2.02; 95% CI: 1.02–4.00; $p$ = 0.044) (Table 2). Another independent risk factor for nonadherence to ART was having a history of gonorrhea infection (AOR = 2.10; 95% CI: 1.01–4.37; $p$ = 0.048). Compared with the HIV-infected MSM with an undetectable viral load, those with a detectable viral load had a lower adherence to ART (AOR = 4.77; 95% CI: 2.38–9.57; $p$ <.001).

Table 3 shows the multivariate analyses for the association between ART adherence and antidepressant therapy in HIV-infected MSM. After controlling for the demographic characteristics and other covariates, it was found that nonadherence to ART was significantly associated with depression without antidepressant therapy (AOR = 4.02; 95% CI: 1.44–11.21; $p$ = 0.008), but that it was not significantly associated with depression with antidepressant therapy (AOR = 1.47; 95% CI: 0.65–3.31; $p$ = 0.358).

## Factors associated with HRQOL among HIV-infected MSM

Table 4 shows the participant characteristics that were associated with each HRQOL domain in the univariate analysis. Physical, psychological, social, and environmental HRQOL were negatively associated with depression and unemployment, but were positively associated with a high income. Use of hypnotic drugs was significantly associated with poor physical and social HRQOL. Moreover, a university education or above was significantly associated with better psychological and environmental HRQOL.

Multiple linear regression analyses were used to evaluate the association between depression and HRQOL in HIV-infected MSM. After controlling for the demographic characteristics and other covariates, depression was found to be significantly associated with poor physical, psychological, social, and environmental HRQOL (Table 5). When antidepressant therapy was considered, the depressed HIV-infected MSM who did not receive antidepressant therapy had worse psychological, social, and environmental HRQOL than the depressed HIV-infected MSM who had antidepressant therapy.

**Table 2. Univariate and multivariate analyses of the factors associated with non-adherence to antiretroviral therapy among HIV-infected MSM.**

| Characteristic | Number of patients | Nonadherence to ART | Univariate analysis | Multivariate analysis |
|---|---|---|---|---|
| | | n (%) | OR (95% CI) | AOR (95% CI) |
| Depression | | | | |
| No | 479 | 52 (10.9) | 1 | 1 |
| Yes | 78 | 17 (21.8) | 2.29 (1.24–4.21)** | 2.02 (1.02–4.00)* |
| **Demographics** | | | | |
| Age, yr | | | | |
| 15–39 | 338 | 50 (14.8) | 1 | 1 |
| ≥40 | 219 | 19 (8.7) | 0.55 (0.31–0.96) | 0.67 (0.36–1.24) |
| Education level completed | | | | |
| ≤High school | 134 | 21 (15.7) | 1 | 1 |
| University or above | 423 | 48 (11.3) | 0.69 (0.40–1.20) | 0.93 (0.50–1.71) |
| Income level | | | | |
| Low | 65 | 11 (16.9) | 1 | 1 |
| Intermediate | 233 | 33 (14.2) | 0.81 (0.38–1.71) | 0.55 (0.20–1.52) |
| High | 259 | 25 (9.7) | 0.52 (0.24–1.13) | 0.48 (0.17–1.36) |
| Unemployment | | | | |
| No | 500 | 62 (12.4) | 1 | 1 |
| Yes | 57 | 7 (12.3) | 0.99 (0.43–2.28) | 0.47 (0.15–1.52) |
| Any alcohol use | | | | |
| No | 304 | 36 (11.8) | 1 | 1 |
| Yes | 253 | 33 (13.0) | 1.12 (0.67–1.85) | 0.84 (0.47–1.48) |
| Smoking | | | | |
| No | 341 | 34 (10.0) | 1 | 1 |
| Yes | 216 | 35 (16.2) | 1.75 (1.05–2.90)* | 1.54 (0.86–2.75) |
| Use of hypnotic drugs | | | | |
| No | 514 | 63 (12.3) | 1 | 1 |
| Yes | 43 | 6 (14.0) | 1.16 (0.47–2.86) | 1.15 (0.44–3.01) |
| **History of sexually transmitted diseases** | | | | |
| History of syphilis | | | | |
| No | 245 | 27 (11.0) | 1 | 1 |
| Yes | 312 | 42 (13.5) | 1.26 (0.75–2.10) | 1.17 (0.68–2.04) |
| History of gonorrhea infection | | | | |
| No | 497 | 56 (11.3) | 1 | 1 |
| Yes | 60 | 13 (21.7) | 2.18 (1.11–4.28)* | 2.10 (1.01–4.37)* |
| History of warts | | | | |
| No | 455 | 54 (11.9) | 1 | 1 |
| Yes | 102 | 15 (14.7) | 1.18 (0.69–2.37) | 1.05 (0.54–2.05) |
| **CD4 count, cells/mm$^3$** | | | | |
| <200 | 74 | 16 (21.6) | 1 | 1 |
| 200–499 | 146 | 22 (15.1) | 0.64 (0.31–1.32) | 0.91 (0.41–2.02) |
| ≥500 | 337 | 31 (9.2) | 0.37 (0.19–0.71)** | 0.51 (0.24–1.08) |
| **HIV-1 RNA, copies/ml** | | | | |
| HIV-1 RNA<40 | 500 | 49 (9.8) | 1 | 1 |
| HIV-1 RNA≥40 | 57 | 20 (35.1) | 4.98 (2.68–9.24)*** | 4.77 (2.38–9.57)*** |

* <.05;

** <.01;

*** <.001

MSM = men who have sex with men; ART = antiretroviral treatment; AOR = adjusted odds ratio; CI = confident interval.

**Table 3. Univariate and multivariate analyses of the association between depression and adherence to antiretroviral therapy among HIV-infected MSM.**

| Characteristic | Number of patients | Nonadherence to ART | Univariate analysis | Multivariate analysis[a] |
|---|---|---|---|---|
| | | n (%) | OR (95% CI) | AOR (95% CI) |
| Depression | | | | |
| No | 479 | 52 (10.9) | 1 | 1 |
| Depression with antidepressant therapy | 55 | 11 (20.0) | 2.05 (1.00–4.22) | 1.47 (0.65–3.31) |
| Depression without antidepressant therapy | 23 | 6 (26.1) | 2.90 (1.09–7.68)* | 4.02 (1.44–11.21)** |

\* <.05;

\*\* <.01;

\*\*\* <.001

[a]after controlling for demographics and history of sexually transmitted diseases.

ART = antiretroviral treatment; AOR = adjusted odds ratio; CI = confident interval.

## Discussion

This study found that 14.0% of the HIV-infected MSM had depression. The proportion of nonadherence to ART was 21.8% and 10.5% in the depressed and nondepressed HIV-infected MSM, respectively. Moreover, physical, psychological, social, and environmental HRQOL in HIV-infected MSM was negatively associated with depression. After adjusting for the demographic characteristics and other covariates, the HIV-infected MSM with depression had poor adherence to ART and poor HRQOL. When the use of antidepressant therapy was considered, the depressed HIV-infected MSM who did not receive antidepressant therapy had poor adherence to ART and worse HRQOL than those who did have antidepressant therapy.

**Table 4. Regression coefficients (standard errors) from the bivariate linear regression analyses of the health-related quality of life among HIV-infected MSM.**

| | HRQOL domain | | | |
|---|---|---|---|---|
| | Physical | Psychological | Social | Environmental |
| **Depression** | −2.14*** (0.29) | −1.90*** (0.34) | −1.58*** (0.31) | −1.46*** (0.28) |
| **Demographics** | | | | |
| Age≥40 years | 0.25 (0.21) | 0.06 (0.25) | −0.13 (0.23) | 0.17 (0.20) |
| Education level: university or above | 0.25 (0.25) | 0.80** (0.28) | 0.40 (0.26) | 0.97*** (0.23) |
| Income level (ref: low) | | | | |
| Intermediate | −0.24 (0.21) | −0.33 (0.25) | −0.02 (0.22) | −0.47* (0.20) |
| High | 0.82*** (0.21) | 0.80*** (0.34) | 0.51* (0.22) | 1.02*** (0.20) |
| Unemployment | −0.68* (0.34) | −1.09** (0.40) | −0.92* (0.36) | −0.85* (0.33) |
| Any alcohol use | −0.03 (0.21) | 0.03 (0.24) | 0.27 (0.22) | 0.10 (0.20) |
| Smoking | −0.20 (0.22) | −0.14 (0.25) | 0.18 (0.23) | −0.05 (0.20) |
| Use of hypnotic drugs | −1.51*** (0.39) | −0.83 (0.45) | −0.84* (0.41) | −0.55 (0.37) |
| **History of sexually transmitted diseases** | | | | |
| History of syphilis | 0.14 (0.21) | 0.40 (0.24) | 0.27 (0.22) | 0.11 (0.20) |
| History of gonorrhea | −0.81* (0.34) | −0.73 (0.39) | −0.47 (0.36) | −0.51 (0.32) |
| History of warts | −0.03 (0.27) | −0.11 (0.31) | 0.18 (0.29) | −0.23 (0.26) |
| **CD4 count, cells/mm$^3$ (ref: <350)** | | | | |
| 350–499 | −0.19 (0.24) | −0.19 (0.28) | −0.15 (0.25) | 0.06 (0.23) |
| ≥500 | 0.19 (0.21) | 0.37 (0.25) | 0.45* (0.23) | 0.35 (0.20) |
| HIV-1 RNA<40 copies/ml | 0.42 (0.35) | 0.33 (0.40) | 0.37 (0.36) | −0.13 (0.33) |

MSM = men who have sex with men; HRQOL = health-related quality of life.

**Table 5. Regression coefficients (standard errors) from the multiple linear regression analyses of the health-related quality of life among HIV-infected MSM.**

|  | HRQOL domain | | | |
|---|---|---|---|---|
|  | **Physical** | **Psychological** | **Social** | **Environmental** |
| **Model I[a]** | | | | |
| Constant | 13.01*** (0.74) | 11.68*** (0.87) | 11.41*** (0.80) | 11.34*** (0.71) |
| Depression | −1.74*** (0.30) | −1.66*** (0.35) | −1.40*** (0.32) | −1.17*** (0.29) |
| Adjusted $R^2$ | 0.12 | 0.07 | 0.06 | 0.1 |
| **Model II (ref: without depression)[a]** | | | | |
| Constant | 13.08*** (0.74) | 11.60*** (0.87) | 11.35*** (0.80) | 11.31*** (0.71) |
| Depression with antidepressant therapy | −1.76*** (0.35) | −1.25** (0.41) | −1.08** (0.38) | −0.89** (0.33) |
| Depression without antidepressant therapy | −1.70*** (0.50) | −2.58*** (0.59) | −2.12*** (0.54) | −1.78*** (0.48) |
| Adjusted $R^2$ | 0.12 | 0.08 | 0.07 | 0.1 |

[a]after controlling for the demographic characteristics and history of sexually transmitted diseases.

MSM = men who have sex with men; HRQOL = health-related quality of life.

This report showed that the prevalence of depression was 14.0% in the Taiwanese HIV-infected MSM, which is higher than 12.4% in HIV-infected individuals in the US [29], but lower than 18% in those of sub-Saharan Africa [30] and 31.3% in HIV-infected MSM in France [31]. The high prevalence of depression in PLWHA may be due to HIV-related biological factors (e.g., alterations in the brain's cortex and subcortex) and psychosocial factors (e.g., HIV stigma and isolation) [18]. Depression in PLWHA has been associated with poor health outcomes, including an impaired immunological response and mortality [20, 32, 33]. However, depressive symptoms in PLWHA are often overlooked in HIV care clinics [5]. Since depression is highly prevalent and associated with poor health outcomes in PLWHA, clinicians should be aware of depressive symptoms in this population.

Our study showed that the depressed HIV-infected MSM had more than twice the risk of poor adherence to ART than the nondepressed HIV-infected MSM. When the use of antidepressant therapy was considered, poor adherence to ART was significantly associated with depression without antidepressant therapy, but it was not significantly related to depression with antidepressant therapy. HIV infection can cause alterations in the cortical and subcortical regions of the brain and may induce the development of depression [18, 34], which could result in the unintentional nonadherence to ART in PLWHA. Although depression is not uncommon in PLWHA [18], routine screening and the treatment of depressive disorders is inadequate in this population [4]. Since depression is associated with poor adherence to ART, our study suggests that it is imperative to screen for depressive symptoms in PLWHA and to provide antidepressant therapy for those with depression.

This study found that the depressed HIV-infected MSM had poor physical, psychological, social, and environmental HRQOL compared to the nondepressed HIV-infected MSM. When considering the use of antidepressant therapy, the depressed HIV-infected MSM without antidepressant therapy had worse psychological, social, and environmental HRQOL than the depressed HIV-infected MSM who were receiving antidepressant therapy. The depressed HIV-infected MSM may have the symptoms of loss of interest, low self-esteem, and psychomotor retardation [18], which can diminish the HRQOL of this population. Although the widespread use of ART has markedly improved survival among PLWHA [35], improving HRQOL is central to their care and support [36]. UNAIDS' HIV testing and treatment targets indicate that 90% of PLWHA who have viral load suppression should have good HRQOL [13, 14]. Since depression is both prevalent and negatively associated with HRQOL in PLWHA,

our study suggests that it is crucial to provide antidepressant therapy for PLWHA who have depression to improve their HRQOL and outcomes.

The present study has several limitations. First, the cross-sectional study design did not allow us to determine the causality between depression and ART adherence and HRQOL. Second, the adherence to ART in study participants were evaluated by using the self-reported MARS-5 rather than testing participants' biological samples for antiretroviral drugs. However, a previous study has reported that MARS-5 has good reliability and validity to evaluate the treatment adherence in patients with chronic diseases [21]. Finally, the external validity of our findings may be of a concern because all participants in our report were PLWHA receiving health care services in clinics. Future studies need to determine the impact of depression on HRQOL in PLWHA not receiving health care services.

In conclusion, this study found that 14.0% of HIV-infected MSM in clinics had depression. After adjusting for the demographic characteristics and other covariates, it was found that HIV-infected MSM with depression had poor adherence to ART and poor HRQOL. MSM infected with HIV and living with depression who did not receive antidepressant therapy were shown to have poor adherence to ART and worse HRQOL compared to those who received antidepressant therapy. To achieve UNAIDS' HIV testing and treatment targets, our study suggests that it is imperative to screen for depressive symptoms in PLWHA and to provide antidepressant therapy for those with depression in order to improve their adherence to ART and their HRQOL.

## Supporting information

**S1 Table. Medication Adherence Report Scale (MARS-5).**
(DOCX)

**S2 Table. Minimal data set of this study.**
(XLS)

## Acknowledgments

The authors gratefully acknowledge Po-Tsen Yeh for interviewing the participants.

## Author Contributions

**Conceptualization:** Yung-Feng Yen, Hsin-Hao Lai, Yen-Chun Kuo, Shang-Yih Chan, Lian-Yu Chen, Chu-Chieh Chen, Teng-Ho Wang, Chien Chun Wang, Marcelo Chen, Tsen-Fang Yen, Li-Lan Kuo, Shu-Ting Kuo, Pei-Hung Chuang.

**Data curation:** Yung-Feng Yen, Hsin-Hao Lai, Yen-Chun Kuo, Shang-Yih Chan, Lian-Yu Chen, Chu-Chieh Chen, Teng-Ho Wang, Chien Chun Wang, Marcelo Chen, Tsen-Fang Yen, Li-Lan Kuo, Shu-Ting Kuo.

**Formal analysis:** Yung-Feng Yen, Shang-Yih Chan, Lian-Yu Chen, Pei-Hung Chuang.

**Funding acquisition:** Yung-Feng Yen.

**Investigation:** Yen-Chun Kuo, Shang-Yih Chan, Lian-Yu Chen, Teng-Ho Wang, Chien Chun Wang, Marcelo Chen, Tsen-Fang Yen, Li-Lan Kuo, Shu-Ting Kuo, Pei-Hung Chuang.

**Methodology:** Yung-Feng Yen, Hsin-Hao Lai, Yen-Chun Kuo, Shang-Yih Chan, Lian-Yu Chen, Chu-Chieh Chen, Teng-Ho Wang, Chien Chun Wang, Marcelo Chen, Tsen-Fang Yen, Li-Lan Kuo, Shu-Ting Kuo, Pei-Hung Chuang.

**Resources:** Yung-Feng Yen, Hsin-Hao Lai, Yen-Chun Kuo, Shang-Yih Chan, Lian-Yu Chen, Chu-Chieh Chen, Teng-Ho Wang, Chien Chun Wang, Marcelo Chen, Tsen-Fang Yen, Li-Lan Kuo, Shu-Ting Kuo, Pei-Hung Chuang.

**Software:** Yung-Feng Yen.

**Supervision:** Yung-Feng Yen.

**Validation:** Yung-Feng Yen.

**Writing – original draft:** Yung-Feng Yen, Hsin-Hao Lai, Yen-Chun Kuo, Shang-Yih Chan, Lian-Yu Chen, Chu-Chieh Chen, Teng-Ho Wang, Chien Chun Wang, Marcelo Chen, Tsen-Fang Yen, Li-Lan Kuo, Shu-Ting Kuo, Pei-Hung Chuang.

**Writing – review & editing:** Yung-Feng Yen, Hsin-Hao Lai, Yen-Chun Kuo, Shang-Yih Chan, Lian-Yu Chen, Chu-Chieh Chen, Teng-Ho Wang, Chien Chun Wang, Marcelo Chen, Tsen-Fang Yen, Li-Lan Kuo, Shu-Ting Kuo, Pei-Hung Chuang.

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
