## [Decision Letter · Decision Letter 0]

18 Oct 2021

PONE-D-21-07210

Association of Depression and Antidepressant Therapy with Antiretroviral Therapy Adherence and Health-Related Quality of Life in Men Who Have Sex with Men

PLOS ONE

Dear Dr. Yen,

Thank you for submitting your manuscript to PLOS ONE. After careful consideration, we feel that it has merit but does not fully meet PLOS ONE’s publication criteria as it currently stands. Therefore, we invite you to submit a revised version of the manuscript that addresses the points raised during the review process.

There are only a few remaining outstanding requests from the peer reviewers. I think the comments are fair and justifiable. Assuming they are all addressed, we should be able to move forward with the manuscript for publication without another round of peer review.

We look forward to receiving your revised manuscript.

Kind regards,

Anthony J. Santella, DrPH, MPH, MCHES

Academic Editor

PLOS ONE

Journal Requirements:

Reviewers' comments:

Reviewer's Responses to Questions

**Comments to the Author**

1. Is the manuscript technically sound, and do the data support the conclusions?

Reviewer #1: Yes

Reviewer #2: Yes

2. Has the statistical analysis been performed appropriately and rigorously? 

Reviewer #1: Yes

Reviewer #2: No

3. Have the authors made all data underlying the findings in their manuscript fully available?

Reviewer #1: Yes

Reviewer #2: Yes

4. Is the manuscript presented in an intelligible fashion and written in standard English?

Reviewer #1: Yes

Reviewer #2: Yes

5. Review Comments to the Author

Reviewer #1: Please find attached up-loaded review with comments and suggestions for manuscript authors. Inputs are provided in the attachment with detailed comments and suggestions. Broader comments include the acknowledgement of additional limitations and a questions regarding MSM/community involvement.

Reviewer #2: This is a very well written manuscript that addresses a topic of great importance in achieving HIV epidemic control. I have attached a couple of minor suggestions but overall great job on documenting the impact of depression on adherence and HRQOL

6. PLOS authors have the option to publish the peer review history of their article (what does this mean?). If published, this will include your full peer review and any attached files.

Reviewer #1: No

Reviewer #2: No

---

## [Author Response · Author response to Decision Letter 0]

24 Oct 2021

October 24, 2021

Anthony J. Santella, DrPH, MPH, MCHES

Editor-in-Chief

PLOS ONE 

Dear Prof. Santella,

We sincerely appreciate the Editor and Reviewers’ detailed comments on our previous manuscript titled, “Association of depression and antidepressant therapy with antiretroviral therapy adherence and health-related quality of life in men who have sex with men”. We also appreciate your invitation to resubmit our revised manuscript to the PLOS ONE. We have carefully reanalyzed our data as recommended and responded to each comment from the Editor and Reviewers. Below, please find our detailed responses. The reviewers’ inputs much improved the manuscript and we look forward to publishing this article in the PLOS ONE.

Please let me know if you have any questions or additional comments.

Sincerely,

Yung-Feng Yen, MD., M.P.H. PhD

Section of Infectious Diseases, Taipei City Hospital, No.145, Zhengzhou Rd., Datong Dist., Taipei City 103, Taiwan (R.O.C.) (e-mail: yfyen1@gmail.com)

 

Reply to the Editor Comments

1: Please ensure that your manuscript meets PLOS ONE's style requirements, including those for file naming. The PLOS ONE style templates can be found at 

Reply 1: We really appreciate your comment. We ensure that our manuscript meets PLOS ONE's style requirements.

2: Please review your reference list to ensure that it is complete and correct. If you have cited papers that have been retracted, please include the rationale for doing so in the manuscript text, or remove these references and replace them with relevant current references. Any changes to the reference list should be mentioned in the rebuttal letter that accompanies your revised manuscript. If you need to cite a retracted article, indicate the article’s retracted status in the References list and also include a citation and full reference for the retraction notice.

Reply 2: We really appreciate your comment. All references list in the manuscript are correct. 

3: In your Data Availability statement, you have not specified where the minimal data set underlying the results described in your manuscript can be found. PLOS defines a study's minimal data set as the underlying data used to reach the conclusions drawn in the manuscript and any additional data required to replicate the reported study findings in their entirety. All PLOS journals require that the minimal data set be made fully available. For more information about our data policy, please see http://journals.plos.org/plosone/s/data-availability.

Reply 3: We really appreciate your comment. We upload out study’s minimal underlying data set as Supporting Information files. 

4: Please include captions for your Supporting Information files at the end of your manuscript, and update any in-text citations to match accordingly. Please see our Supporting Information guidelines for more information: http://journals.plos.org/plosone/s/supporting-information

Reply 4: We really appreciate your comment. We include captions for our supporting Information files at the end of our manuscript “

Supplementary Table 1. Medication Adherence Report Scale (MARS-5).

Supplementary Table 2. Minimal data set of this study” Please see Page 21.

 

Reply to the Reviewer 1 Comments

Broader comments:

1: Participants enrolled include only MSM already enrolled in HIV and health care services, thus not capturing mental health issues or depression among MSM not accessing health facilities. A recommendation for further study of depression could be made that facility-based data should be complemented by data form a survey such as an MSM IBBS that would collect information also from those not accessing facility-based services. It is possible that rates of depression might be higher or even higher among those not accessing services.

Reply 1: We really appreciate your comment. All participants in our report were PLWHA receiving health care services in clinics, which may cause a concern of external validity. Future studies need to determine the impact of depression on HRQOL in PLWHA not receiving health care services. We add this information in the limitation section “Finally, the external validity of our findings may be of a concern because all participants in our report were PLWHA receiving health care services in clinics. Future studies need to determine the impact of depression on HRQOL in PLWHA not receiving health care services.” Please see Page 16 (first paragraph, line 3-6).

2: The scope and/or funding for this study may not have allowed for this, it would however have been best to collect and test biological samples for ARVs to assess ART adherence rather than to rely on self-reported behaviors.

Reply 2: We really appreciate your comment. The adherence to ART in study participants were evaluated by using the self-reported MARS-5 rather than testing participants’ biological samples for antiretroviral drugs. However, a previous study has reported that MARS-5 has good reliability and validity to evaluate the treatment adherence in patients with chronic diseases. We add this information in the limitation section “Second, the adherence to ART in study participants were evaluated by using the self-reported MARS-5 rather than testing participants’ biological samples for antiretroviral drugs. However, a previous study has reported that MARS-5 has good reliability and validity to evaluate the treatment adherence in patients with chronic diseases [21].” Please see Page 16 (third paragraph, line 3-6).

 

Detailed comments:

1: Line 77-78: This would be true in the absence of other prevention measures being used, e.g. correct and consistent condom use, HIV-negative partner(s) on PrEP.

Reply 1: We really appreciate your comment. We revise the sentence in line 77-78 as “Poor adherence to ART in PLWHA not only increases the risk of death [9] but also results in the transmission of HIV to others in the absence of condom usage or pre-exposure prophylaxis [10].” Please see Page 5 (second paragraph, line 7-9).

2: Line 83-84: Somewhat misquotes the new UNAIDS targets. This should correctly quote the new UNAIDS 2025, with the only relating or covering this use as “90% of People Living with HIV and People at Risk are Linked to Other Integrated Health Services”, which mental health services would be part of. (Source: “End Inequalities. End AIDS. Global AIDS Strategy 2021-2026”, UNAIDS 2021)

Reply 2: We really appreciate your comment. We revise the statement in line 83-84 as “UNAIDS has added a “fourth 90” to the prior three-pronged 90-90-90 targets to improve mental well-being in PLWHA, which ensures that 90% of PLWHA have good HRQOL and are linked to integrated health services [12-15].” Please see Page 5 (third paragraph, line 5-8).

3: Line 113-114: Sentence is unclear and suggest that enrolment has not been limited to MSM but included other PLWHA. While this is clarified in the method section later (stating only MS were eligible), it should be clarified or corrected here also.

Reply 3: We really appreciate your comment. Since the numbers of heterosexuals infected with HIV (n=27) and PLWHA who were female who had sex with female (n=1) are limited, this study only included PLWHA in the analysis who were men who had sex with men (n=565) to determine the association of depression and antidepressant therapy with ART adherence and HRQOL. We add this information in the method “Since the numbers of heterosexuals infected with HIV (n=27) and PLWHA who were female who had sex with female (n=1) are limited, this study only included PLWHA in the analysis who were men who had sex with men (n=565) to determine the association of depression and antidepressant therapy with ART adherence and HRQOL.” Please see Page 7 (second paragraph, line 1-4).

4: Line 136-137: Appreciate the adaptation and addition of items specific to Taiwanese culture. Could you kindly provide some background to the item “eating what one likes to eat” to the environmental domain, given many readers may not be familiar with the Taiwanese culture?

Reply 4: We really appreciate your comment. This study used Taiwanese version of WHOQOL-BREF to evaluate the participants’ HRQOL, which includes the 26 original items of the WHOQOL-BREF plus two culture-specific items that are relevant to Taiwan. These two culture-specific items were proposed by patient and expert focus groups after a qualitative analysis of the recorded content. Based on psychometric analyses, one culture-specific item that addresses “respect from others” was included in the social domain, and the other item that corresponds to “eating what one likes to eat” was included in the environmental domain. We add this information in the method “The participants’ HRQOL was evaluated using the Taiwanese version of the short form of the World Health Organization Quality of Life questionnaire (WHOQOL-BREF). The Taiwanese version includes the 26 original items of the WHOQOL-BREF [25] plus two culture-specific items that are relevant to Taiwan [26], that were proposed by patient and expert focus groups after a qualitative analysis of the recorded content. Based on the psychometric analyses, one culture-specific item that addresses “respect from others” was included in the social domain, and the other item that corresponds to “eating what one likes to eat” was included in the environmental domain.” Please see Page 8 (second paragraph, line 1-8).

5: Line 194-196: Please add AOR, CI and p-value for these associations, i.e. lower income, drug use, gonorrhea.

Reply 5: We really appreciate your comment. We add AOR, CI and p-value for gonorrhea and viral load in the result “Another independent risk factor for nonadherence to ART was having a history of gonorrhea infection (AOR = 2.10; 95% CI: 1.01-4.37; p = 0.048). Compared with the HIV-infected MSM with an undetectable viral load, those with a detectable viral load had a lower adherence to ART (AOR = 4.77; 95% CI: 2.38-9.57; p <.001).” Please see Page 11 (third paragraph, line 4-8).

6: Line 238-239: Date are provided on depression from the US, SSA and France. What are baseline data or estimates for Taiwan or for the Asia region as more relevant (if known)?

Reply 6: We really appreciate your comment. The prevalence of depression in the whole HIV-infected MSM in Taiwan was not available. Our study found that the prevalence of depression was 14.0% in HIV-infected MSM in the largest HIV care center in Taiwan, which is higher than 12.4% in HIV-infected individuals in the US, but lower than 18% in HIV-infected individuals in sub-Saharan Africa and 31.3% in HIV-infected MSM in France. We add this information in the discussion section “This report showed that the prevalence of depression was 14.0% in the Taiwanese HIV-infected MSM, which is higher than 12.4% in HIV-infected individuals in the US [29], but lower than 18% in those of sub-Saharan Africa [30] and 31.3% in HIV-infected MSM in France [31].” Please see Page 14 (second paragraph, line 1-4).

7: Line 276: First sentence in the conclusion section should specify or add that this applies for clinical sites in Taiwan, where MSM were accessing services.

Reply 7: We really appreciate your comment. We revise the first sentence in the conclusion as “this study found that 14.0% of HIV-infected MSM in clinics had depression.” Please see Page 16 (second paragraph, line 1).

 

Reply to the Reviewer 2 Comments

Introduction:

1: The authors might want to correct the interpretation of the 2nd in UNAIDS 90-90-90 to “sustained ART” instead of started ART (line 72-73).

Reply 1: We really appreciate your comment. We revise the interpretation of the 2nd in UNAIDS 90-90-90 as “90% of those aware have sustained ART”. Please see Page 5 (second paragraph, line 4).

Method:

1: Methods address the study goals and as a strength, the authors added some culturally relevant questions to the HRQOL measures. The authors do not describe the total number of MSM in the clinics and how the samples size was arrived at or was adequate.

Reply 1: We really appreciate your comment. As of the end of 2019, a total of 4,150 PLWHA are regularly followed-up at Taipei City Hospital (TCH) HIV clinics. However, the proportion of HIV-infected MSM in all PLWHA at TCH HIV clinics is not available. This study consecutively recruited HIV-infected patients from TCH HIV clinics between December 2018 and May 2020. We add this information in the method “This study consecutively recruited HIV-infected patients from Taipei City Hospital (TCH) HIV clinics, the largest HIV care center in Taiwan, between December 2018 and May 2020. As of the end of 2019, a total of 4,150 PLWHA are regularly followed-up at TCH HIV clinics.” Please see Page 7 (first paragraph, line 1-3).

Results:

1: 565 MSM enrolled but not clear if this was a representative sample for MSM in the clinic. Was this an adequate sample? See comment in the methods section.

Reply 1: We really appreciate your comment. As of the end of 2019, a total of 4,150 PLWHA are regularly followed-up at Taipei City Hospital (TCH) HIV clinics. However, the proportion of HIV-infected MSM in all PLWHA at TCH HIV clinics is not available. This study consecutively recruited HIV-infected patients from TCH HIV clinics between December 2018 and May 2020. We add this information in the method “This study consecutively recruited HIV-infected patients from Taipei City Hospital (TCH) HIV clinics, the largest HIV care center in Taiwan, between December 2018 and May 2020. As of the end of 2019, a total of 4,150 PLWHA are regularly followed-up at TCH HIV clinics.” Please see Page 7 (first paragraph, line 1-3).

2: Line 194. Small typo. Use “terms” instead of “term”

Reply 2: We really appreciate your comment. We revise “term” as “terms” in line 201-202 “In terms of the sexually transmitted diseases…”Please see Page 11 (second paragraph, line 5).

3: Line 200. After controlling “for” instead of controlling the

Reply 3: We really appreciate your comment. We revise “After controlling the demographic characteristics…” as “After controlling for the demographic characteristics…” in the result section. Please see Page 11 (third paragraph).

4: Line 221. Use controlling “for”

Reply 4: We really appreciate your comment. We revise “After controlling the demographic characteristics…” as “After controlling for the demographic characteristics…” in the result section. Please see Page 12 (second paragraph, line 2).

Conclusion:

1: Results should not be repeated in the conclusion verbatim. Line 278-281 should be rephrased since it appears in the results section almost in the same way.

Reply 1: We really appreciate your comment. We revise the sentence in line 278-281 in the conclusion as “MSM infected with HIV and living with depression who did not receive antidepressant therapy were shown to have poor adherence to ART and worse HRQOL compared to those who received antidepressant therapy.” Please see Page 16 (second paragraph, line 3-6).

---

## [Decision Letter · Decision Letter 1]

14 Feb 2022

Association of depression and antidepressant therapy with antiretroviral therapy adherence and health-related quality of life in men who have sex with men

PONE-D-21-07210R1

Dear Dr. Yen,

We’re pleased to inform you that your manuscript has been judged scientifically suitable for publication and will be formally accepted for publication once it meets all outstanding technical requirements.

Kind regards,

Anthony J. Santella, DrPH, MPH, MCHES

Academic Editor

PLOS ONE

Additional Editor Comments (optional):

Reviewers' comments:

Reviewer's Responses to Questions

**Comments to the Author**

1. If the authors have adequately addressed your comments raised in a previous round of review and you feel that this manuscript is now acceptable for publication, you may indicate that here to bypass the “Comments to the Author” section, enter your conflict of interest statement in the “Confidential to Editor” section, and submit your "Accept" recommendation.

Reviewer #2: All comments have been addressed

Reviewer #3: All comments have been addressed

2. Is the manuscript technically sound, and do the data support the conclusions?

Reviewer #2: Yes

Reviewer #3: Yes

3. Has the statistical analysis been performed appropriately and rigorously? 

Reviewer #2: I Don't Know

Reviewer #3: Yes

4. Have the authors made all data underlying the findings in their manuscript fully available?

Reviewer #2: Yes

Reviewer #3: Yes

5. Is the manuscript presented in an intelligible fashion and written in standard English?

Reviewer #2: Yes

Reviewer #3: Yes

6. Review Comments to the Author

Reviewer #2: This is a much improved and well written manuscript with detailed responses to mine and other reviewer comments. The paper reads very well and will contribute to the body of evidence supporting the achievement of optimal outcomes for people living with HIV.

Reviewer #3: Dear author

I believe that, this is a professional article with high quality of meta-analysis. I think, it can have great impact on health sciences and also it can get a lot of citations which will be beneficial. Thanks for your answer. I think all comments have been correctly answered.

7. PLOS authors have the option to publish the peer review history of their article (what does this mean?). If published, this will include your full peer review and any attached files.

Reviewer #2: **Yes: **Moses H Bateganya

Reviewer #3: No

---

## [Editor Report · Acceptance letter]

16 Feb 2022

PONE-D-21-07210R1 

Association of depression and antidepressant therapy with antiretroviral therapy adherence and health-related quality of life in men who have sex with men 

Dear Dr. Yen:

I'm pleased to inform you that your manuscript has been deemed suitable for publication in PLOS ONE. Congratulations! Your manuscript is now with our production department. 

Kind regards, 

on behalf of

Dr. Anthony J. Santella 

Academic Editor

PLOS ONE